# Split-$\mathrm{kl}$ and PAC-Bayes-split-$\mathrm{kl}$ Inequalities for Ternary Random Variables

**Yi-Shan Wu**
University of Copenhagen
yswu@di.ku.dk

**Yevgeny Seldin**
University of Copenhagen
seldin@di.ku.dk

## Abstract

We present a new concentration of measure inequality for sums of independent bounded random variables, which we name a split-$\mathrm{kl}$ inequality. The inequality is particularly well-suited for ternary random variables, which naturally show up in a variety of problems, including analysis of excess losses in classification, analysis of weighted majority votes, and learning with abstention. We demonstrate that for ternary random variables the inequality is simultaneously competitive with the $\mathrm{kl}$ inequality, the Empirical Bernstein inequality, and the Unexpected Bernstein inequality, and in certain regimes outperforms all of them. It resolves an open question by Tolstikhin and Seldin [2013] and Mhammedi et al. [2019] on how to match simultaneously the combinatorial power of the $\mathrm{kl}$ inequality when the distribution happens to be close to binary and the power of Bersntein inequalities to exploit low variance when the probability mass is concentrated on the middle value. We also derive a PAC-Bayes-split-$\mathrm{kl}$ inequality and compare it with the PAC-Bayes-$\mathrm{kl}$, PAC-Bayes-Empirical-Bennett, and PAC-Bayes-Unexpected-Bernstein inequalities in an analysis of excess losses and in an analysis of a weighted majority vote for several UCI datasets. Last, but not least, our study provides the first direct comparison of the Empirical Bernstein and Unexpected Bernstein inequalities and their PAC-Bayes extensions.

## 1 Introduction

Concentration of measure inequalities for sums of independent random variables are the most fundamental analysis tools in statistics and many other domains [Boucheron et al., 2013]. Their history stretches almost a century back, and inequalities such as Hoeffding's [Hoeffding, 1963] and Bernstein's [Bernstein, 1946] are the main work horses of learning theory.

For binary random variables, one of the tightest concentration of measure inequalities is the $\mathrm{kl}$ inequality [Maurer, 2004, Langford, 2005, Foong et al., 2021, 2022], which is based on combinatorial properties of a sum of $n$ independent random variables.[1] However, while being extremely tight for binary random variables and applicable to any bounded random variables, the $\mathrm{kl}$ inequality is not necessarily a good choice for sums of bounded random variables that can take more than two values. In the latter case, the Empirical Bernstein [Mnih et al., 2008, Audibert et al., 2009, Maurer and Pontil, 2009] and the Unexpected Bernstein [Cesa-Bianchi et al., 2007, Mhammedi et al., 2019] inequalities can be significantly tighter due to their ability to exploit low variance, as shown by Tolstikhin and Seldin [2013]. However, the Empirical and Unexpected Bernstein inequalities are loose for binary random variables [Tolstikhin and Seldin, 2013].

---

[1]The Binomial tail bound is slightly tighter, but it does not extend to the PAC-Bayes setting [Langford, 2005]. Our split-$\mathrm{kl}$ approach can be directly applied to obtain a "split-Binomial-tail" inequality.

36th Conference on Neural Information Processing Systems (NeurIPS 2022).

The challenge of exploiting low variance and, at the same time, matching the tightness of the kl inequality if a distribution happens to be close to binary, was faced by multiple prior works [Tolstikhin and Seldin, 2013, Mhammedi et al., 2019, Wu et al., 2021], but remained an open question. We resolve this question for the case of ternary random variables. Such random variables appear in a variety of applications, and we illustrate two of them. One is a study of excess losses, which are differences between the zero-one losses of a prediction rule $h$ and a reference prediction rule $h^*$, $Z = \ell(h(X), Y) - \ell(h^*(X), Y) \in \{-1, 0, 1\}$. Mhammedi et al. [2019] have applied the PAC-Bayes-Unexpected-Bernstein bound to excess losses in order to improve generalization bounds for classification. Another example of ternary random variables is the tandem loss with an offset, defined by $\ell_\alpha(h(X), h'(X), Y) = (\ell(h(X), Y) - \alpha)(\ell(h'(X), Y) - \alpha) \in \{\alpha^2, -\alpha(1-\alpha), (1-\alpha)^2\}$. Wu et al. [2021] have applied the PAC-Bayes-Empirical-Bennett inequality to the tandem loss with an offset to obtain a generalization bound for the weighted majority vote. Yet another potential application, which we leave for future work, is learning with abstention [Cortes et al., 2018, Thulasidasan et al., 2019].

We present the split-kl inequality, which simultaneously matches the tightness of the Empirical/Unexpected Bernstein and the kl, and outperforms both for certain distributions. It works for sums of any bounded random variables $Z_1, \ldots, Z_n$, not only the ternary ones, but it is best suited for ternary random variables, for which it is almost tight (in the same sense, as the kl is tight for binary random variables). The idea behind the split-kl inequality is to write a random variable $Z$ as $Z = \mu + Z^+ - Z^-$, where $\mu$ is a constant, $Z^+ = \max\{0, Z - \mu\}$, and $Z^- = \max\{0, \mu - Z\}$. Then $\mathbb{E}[Z] = \mu + \mathbb{E}[Z^+] - \mathbb{E}[Z^-]$ and, given an i.i.d. sample $Z_1, \ldots, Z_n$, we can bound the distance between $\frac{1}{n}\sum_{i=1}^n Z_i$ and $\mathbb{E}[Z]$ by using kl upper and lower bounds on the distances between $\frac{1}{n}\sum_{i=1}^n Z_i^+$ and $\mathbb{E}[Z^+]$, and $\frac{1}{n}\sum_{i=1}^n Z_i^-$ and $\mathbb{E}[Z^-]$, respectively. For ternary random variables $Z \in \{a, b, c\}$ with $a \leq b \leq c$, the best split is to take $\mu = b$, then both $Z^+$ and $Z^-$ are binary and the kl upper and lower bounds for their rescaled versions are tight and, therefore, the split-kl inequality for $Z$ is also tight. Thus, this approach provides the best of both worlds: the combinatorial tightness of the kl bound and exploitation of low variance when the probability mass on the middle value happens to be large, as in Empirical Bernstein inequalities. We further elevate the idea to the PAC-Bayes domain and derive a PAC-Bayes-split-kl inequality.

We present an extensive set of experiments, where we first compare the kl, Empirical Bernstein, Unexpected Bernstein, and split-kl inequalities applied to (individual) sums of independent random variables in simulated data, and then compare the PAC-Bayes-kl, PAC-Bayes-Unexpected-Bersnstein, PAC-Bayes-split-kl, and, in some of the setups, PAC-Bayes-Empirical-Bennett, for several prediction models on several UCI datasets. In particular, we evaluate the bounds in the linear classification setup studied by Mhammedi et al. [2019] and in the weighted majority prediction setup studied by Wu et al. [2021]. To the best of our knowledge, this is also the first time when the Empirical Bernstein and the Unexpected Bernstein inequalities are directly compared, with and without the PAC-Bayesian extension. In Appendix A.2 we also show that an inequality introduced by Cesa-Bianchi et al. [2007] yields a relaxation of the Unexpected Bernstein inequality by Mhammedi et al. [2019].

## 2 Concentration of Measure Inequalities for Sums of Independent Random Variables

We start with the most basic question in probability theory and statistics: how far can an average of an i.i.d. sample $Z_1, \ldots, Z_n$ deviate from its expectation? We cite the major existing inequalities, the kl, Empirical Bernstein, and Unexpected Bernstein, then derive the new split-kl inequality, and then provide a numerical comparison.

### 2.1 Background

We use $\mathrm{KL}(\rho\|\pi)$ to denote the Kullback-Leibler divergence between two probability distributions, $\rho$ and $\pi$ [Cover and Thomas, 2006]. We further use $\mathrm{kl}(p\|q)$ as a shorthand for the Kullback-Leibler divergence between two Bernoulli distributions with biases $p$ and $q$, namely $\mathrm{kl}(p\|q) = \mathrm{KL}((1-p, p)\|(1-q, q))$. For $\hat{p} \in [0, 1]$ and $\varepsilon \geq 0$ we define the upper and lower inverse of kl, respectively, as $\mathrm{kl}^{-1,+}(\hat{p}, \varepsilon) := \max\{p : p \in [0, 1] \text{ and } \mathrm{kl}(\hat{p}\|p) \leq \varepsilon\}$ and $\mathrm{kl}^{-1,-}(\hat{p}, \varepsilon) := \min\{p : p \in [0, 1] \text{ and } \mathrm{kl}(\hat{p}\|p) \leq \varepsilon\}$.

The first inequality that we cite is the kl inequality.

**Theorem 1** (kl Inequality [Langford, 2005, Foong et al., 2021, 2022]). *Let $Z_1, \cdots, Z_n$ be i.i.d. random variables bounded in the $[0,1]$ interval and with $\mathbb{E}[Z_i] = p$ for all $i$. Let $\hat{p} = \frac{1}{n}\sum_{i=1}^{n} Z_i$ be their empirical mean. Then, for any $\delta \in (0,1)$:*

$$\mathbb{P}\left( \mathrm{kl}(\hat{p}\|p) \geq \frac{\ln\frac{1}{\delta}}{n} \right) \leq \delta$$

*and, by inversion of the* kl,

$$\mathbb{P}\left( p \geq \mathrm{kl}^{-1,+}\left( \hat{p}, \frac{1}{n}\ln\frac{1}{\delta} \right) \right) \leq \delta, \tag{1}$$

$$\mathbb{P}\left( p \leq \mathrm{kl}^{-1,-}\left( \hat{p}, \frac{1}{n}\ln\frac{1}{\delta} \right) \right) \leq \delta. \tag{2}$$

We note that the PAC-Bayes-kl inequality (Theorem 5 below) is based on the inequality $\mathbb{E}\left[ e^{n\,\mathrm{kl}(\hat{p}\|p)} \right] \leq 2\sqrt{n}$ [Maurer, 2004], which gives $\mathbb{P}\left( \mathrm{kl}(\hat{p}\|p) \geq \frac{\ln\frac{2\sqrt{n}}{\delta}}{n} \right) \leq \delta$. Foong et al. [2021, 2022] reduce the logarithmic factor down to $\ln\frac{1}{\delta}$ by basing the proof on Chernoff's inequality, but this proof technique cannot be combined with PAC-Bayes. Therefore, when we move on to PAC-Bayes we pay the extra $\ln 2\sqrt{n}$ factor in the bounds. It is a long-standing open question whether this factor can be reduced in the PAC-Bayesian setting [Foong et al., 2021].

Next we cite two versions of the Empirical Bernstein inequality.

**Theorem 2** (Empirical Bernstein Inequality [Maurer and Pontil, 2009]). *Let $Z_1, \cdots, Z_n$ be i.i.d. random variables bounded in a $[a,b]$ interval for some $a, b \in \mathbb{R}$, and with $\mathbb{E}[Z_i] = p$ for all $i$. Let $\hat{p} = \frac{1}{n}\sum_{i=1}^{n} Z_i$ be the empirical mean and let $\hat{\sigma} = \frac{1}{n-1}\sum_{i=1}^{n}(Z_i - \hat{p})^2$ be the empirical variance. Then for any $\delta \in (0,1)$ :*

$$\mathbb{P}\left( p \geq \hat{p} + \sqrt{\frac{2\hat{\sigma}\ln\frac{2}{\delta}}{n}} + \frac{7(b-a)\ln\frac{2}{\delta}}{3(n-1)} \right) \leq \delta. \tag{3}$$

**Theorem 3** (Unexpected Bernstein Inequality [Fan et al., 2015, Mhammedi et al., 2019]). *Let $Z_1, \cdots, Z_n$ be i.i.d. random variables bounded from above by $b$ for some $b > 0$, and with $\mathbb{E}[Z_i] = p$ for all $i$. Let $\hat{p} = \frac{1}{n}\sum_{i=1}^{n} Z_i$ be the empirical mean and let $\hat{\sigma} = \frac{1}{n}\sum_{i=1}^{n} Z_i^2$ be the empirical mean of the second moments. Let $\psi(u) := u - \ln(1+u)$ for $u > -1$. Then, for any $\gamma \in (0, 1/b)$ and any $\delta \in (0,1)$:*

$$\mathbb{P}\left( p \geq \hat{p} + \frac{\psi(-\gamma b)}{\gamma b^2}\hat{\sigma} + \frac{\ln\frac{1}{\delta}}{\gamma n} \right) \leq \delta. \tag{4}$$

To facilitate a comparison with other bounds, Theorem 3 provides a slightly different form of the Unexpected Bernstein inequality than the one used by Mhammedi et al. [2019]. We provide a proof of the theorem in Appendix A.1, which is based on the Unexpected Bernstein Lemma [Fan et al., 2015]. We note that an inequality proposed by Cesa-Bianchi et al. [2007] can be used to derive a relaxed version of the Unexpected Bernstein inequality, as discussed in Appendix A.2.

## 2.2 The Split-kl Inequality

Let $Z$ be a random variable bounded in a $[a,b]$ interval for some $a, b \in \mathbb{R}$ and let $\mu \in [a,b]$ be a constant. We decompose $Z = \mu + Z^+ - Z^-$, where $Z^+ = \max(0, Z - \mu)$ and $Z^- = \max(0, \mu - Z)$. Let $p = \mathbb{E}[Z]$, $p^+ = \mathbb{E}[Z^+]$, and $p^- = \mathbb{E}[Z^-]$. For an i.i.d. sample $Z_1, \ldots, Z_n$ let $\hat{p}^+ = \frac{1}{n}\sum_{i=1}^{n} Z_i^+$ and $\hat{p}^- = \frac{1}{n}\sum_{i=1}^{n} Z_i^-$.

With these definitions we present the split-kl inequality.

**Theorem 4** (Split-kl inequality). *Let $Z_1, \ldots, Z_n$ be i.i.d. random variables in a $[a,b]$ interval for some $a, b \in \mathbb{R}$, then for any $\mu \in [a,b]$ and $\delta \in (0,1)$:*

$$\mathbb{P}\left( p \geq \mu + (b-\mu)\,\mathrm{kl}^{-1,+}\left( \frac{\hat{p}^+}{b-\mu}, \frac{1}{n}\ln\frac{2}{\delta} \right) - (\mu-a)\,\mathrm{kl}^{-1,-}\left( \frac{\hat{p}^-}{\mu-a}, \frac{1}{n}\ln\frac{2}{\delta} \right) \right) \leq \delta. \tag{5}$$

*Proof.*

$$\mathbb{P}\left(p \geq \mu + (b-\mu)\,\mathrm{kl}^{-1,+}\left(\frac{\hat{p}^+}{b-\mu}, \frac{1}{n}\ln\frac{2}{\delta}\right) - (\mu-a)\,\mathrm{kl}^{-1,-}\left(\frac{\hat{p}^-}{\mu-a}, \frac{1}{n}\ln\frac{2}{\delta}\right)\right)$$

$$\leq \mathbb{P}\left(p^+ \geq (b-\mu)\,\mathrm{kl}^{-1,+}\left(\frac{\hat{p}^+}{b-\mu}, \frac{1}{n}\ln\frac{2}{\delta}\right)\right) + \mathbb{P}\left(p^- \leq (\mu-a)\,\mathrm{kl}^{-1,-}\left(\frac{\hat{p}^-}{\mu-a}, \frac{1}{n}\ln\frac{2}{\delta}\right)\right)$$

$$\leq \delta,$$

where the last inequality follows by application of the kl upper and lower bounds from Theorem 1 to the first and second terms in the middle line, respectively. ∎

For ternary random variables the best choice is to take $\mu$ to be the middle value, then the resulting $Z^+$ and $Z^-$ are binary and the corresponding kl upper and lower bounds on $p^+$ and $p^-$ are tight, and the resulting split-kl bound is tight. The inequality can be applied to any bounded random variables, but same way as the kl inequality is not necessarily a good choice for bounded random variables, if the distribution is not binary, the split-kl in not necessarily a good choice if the distribution is not ternary.

### 2.3 Empirical Comparison

We present an empirical comparison of the tightness of the above four concentration inequalities: the kl, the Empirical Bernstein, the Unexpected Bernstein, and the split-kl. We take $n$ i.i.d. samples $Z_1, \ldots, Z_n$ taking values in $\{-1, 0, 1\}$. The choice is motivated both by instructiveness of presentation and by subsequent applications to excess losses. We let $p_{-1} = \mathbb{P}(Z = -1)$, $p_0 = \mathbb{P}(Z = 0)$, and $p_1 = \mathbb{P}(Z = 1)$, where $p_{-1} + p_0 + p_1 = 1$. Then $p = \mathbb{E}[Z] = p_1 - p_{-1}$. We also let $\hat{p} = \frac{1}{n}\sum_{i=1}^{n} Z_i$.

In Figure 1 we plot the difference between the bounds on $p$ given by the inequalities (1), (3), (4), and (5), and $\hat{p}$. Lower values in the plot correspond to tighter bounds. To compute the kl bound we first rescale the losses to the $[0, 1]$ interval, and then rescale the bound back to the $[-1, 1]$ interval. For the Empirical Bernstein bound we take $a = -1$ and $b = 1$. For the Unexpected Bernstein bound we take a grid of $\gamma \in \{1/(2b), \cdots, 1/(2^k b)\}$ for $k = \lceil \log_2(\sqrt{n/\ln(1/\delta)}/2)\rceil$ and a union bound over the grid, as proposed by Mhammedi et al. [2019]. For the split-kl bound we take $\mu$ to be the middle value, 0, of the ternary random variable. In the experiments we take $\delta = 0.05$, and truncate the bounds at 1.

In the first experiment, presented in Figure 1a, we take $p_{-1} = p_1 = (1-p_0)/2$ and plot the difference between the values of the bounds and $\hat{p}$ as a function of $p_0$. For $p_0 = 0$ the random variable $Z$ is Bernoulli and, as expected, the kl inequality performs the best, followed by split-kl, and then Unexpected Bernstein. As $p_0$ grows closer to 1, the variance of $Z$ decreases and, also as expected, the kl inequality falls behind, whereas split-kl and Unexpected Bernstein go closely together. Empirical Bernstein falls behind all other bounds throughout most of the range, except slightly outperforming kl when $p_0$ gets very close to 1.

In the second experiment, presented in Figure 1b, we take a skewed random variable with $p_1 = 0.99(1-p_0)$ and $p_{-1} = 0.01(1-p_0)$, and again plot the difference between the values of the bounds and $\hat{p}$ as a function of $p_0$. This time the kl also starts well for $p_0$ close to zero, but then falls behind due to its inability of properly handling the values inside the interval. Unexpected Bernstein exhibits the opposite trend due to being based on uncentered second moment, which is high when $p_0$ is close to zero, even though the variance is small in this case. Empirical Bernstein lags behind all other bounds for most of the range due to poor constants, whereas split-kl matches the tightest bounds, the kl and Unexpected Bernstein, at the endpoints of the range of $p_0$, and outperforms all other bounds in the middle of the range, around $p_0 = 0.6$, due to being able to exploit the combinatorics of the problem.

The experiments demonstrate that for ternary random variables the split-kl is a powerful alternative to existing concentration of measure inequalities. To the best of our knowledge, this is also the first empirical evaluation of the Unexpected Bernstein inequality, and it shows that in many cases it is also a powerful inequality. We also observe that in most settings the Empirical Bernstein is weaker than the other three inequalities we consider. Numerical evaluations in additional settings are provided in Appendix D.

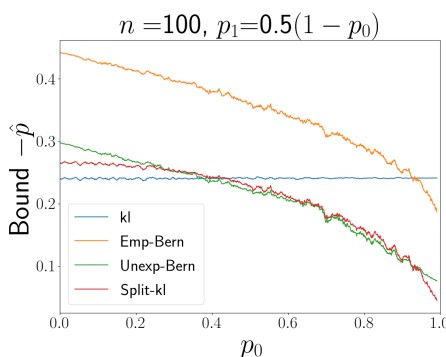

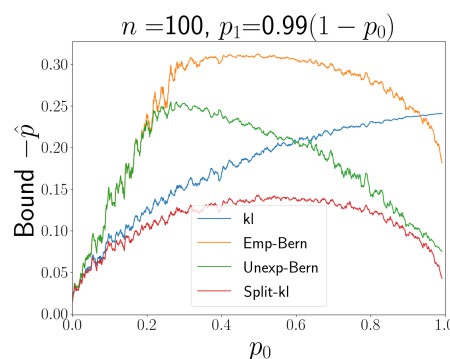

(a) Comparison of the concentration bounds with $n = 100$, $\delta = 0.05$ and $p_{-1} = p_1 = 0.5(1 - p_0)$.

(b) Comparison of the concentration bounds with $n = 100$, $\delta = 0.05$, $p_1 = 0.99(1 - p_0)$, and $p_{-1} = 0.01(1 - p_0)$.

Figure 1: Empirical comparison of the concentration bounds.

## 3 PAC-Bayesian Inequalities

Now we elevate the basic concentration of measure inequalities to the PAC-Bayesian domain. We start with the supervised learning problem setup, then provide a background on existing PAC-Bayesian inequalities, and finish with presentation of the PAC-Bayes-split-kl inequality.

### 3.1 Supervised Learning Problem Setup and Notations

Let $\mathcal{X}$ be a sample space, $\mathcal{Y}$ be a label space, and let $S = \{(X_i, Y_i)\}_{i=1}^n$ be an i.i.d. sample drawn according to an unknown distribution $\mathcal{D}$ on the product-space $\mathcal{X} \times \mathcal{Y}$. Let $\mathcal{H}$ be a hypothesis space containing hypotheses $h : \mathcal{X} \to \mathcal{Y}$. The quality of a hypothesis $h$ is measured using the zero-one loss $\ell(h(X), Y) = \mathbb{1}(h(X) \neq Y)$, where $\mathbb{1}(\cdot)$ is the indicator function. The expected loss of $h$ is denoted by $L(h) = \mathbb{E}_{(X,Y)\sim\mathcal{D}}[\ell(h(X), Y)]$, and the empirical loss of $h$ on a sample $S$ is denoted by $\hat{L}(h, S) = \frac{1}{|S|} \sum_{(X,Y)\in S} \ell(h(X), Y)$. We use $\mathbb{E}_{\mathcal{D}}[\cdot]$ as a shorthand for $\mathbb{E}_{(X,Y)\sim\mathcal{D}}[\cdot]$.

PAC-Bayesian bounds bound the generalization error of Gibbs prediction rules. For each input $X \in \mathcal{X}$, Gibbs prediction rule associated with a distribution $\rho$ on $\mathcal{H}$ randomly draws a hypothesis $h \in \mathcal{H}$ according to $\rho$ and predicts $h(X)$. The expected loss of the Gibbs prediction rule is $\mathbb{E}_{h\sim\rho}[L(h)]$ and the empirical loss is $\mathbb{E}_{h\sim\rho}[\hat{L}(h, S)]$. We use $\mathbb{E}_{\rho}[\cdot]$ as a shorthand for $\mathbb{E}_{h\sim\rho}[\cdot]$.

### 3.2 PAC-Bayesian Analysis Background

Now we present a brief background on the relevant results from the PAC-Bayesian analysis.

**PAC-Bayes-kl Inequality** The PAC-Bayes-kl inequality cited below is one of the tightest known generalization bounds on the expected loss of the Gibbs prediction rule.

**Theorem 5** (PAC-Bayes-kl Inequality, Seeger, 2002, Maurer, 2004). *For any probability distribution $\pi$ on $\mathcal{H}$ that is independent of $S$ and any $\delta \in (0, 1)$:*

$$\mathbb{P}\left( \exists \rho \in \mathcal{P} : \mathrm{kl}\left( \mathbb{E}_{\rho}[\hat{L}(h, S)] \middle\| \mathbb{E}_{\rho}[L(h)] \right) \geq \frac{\mathrm{KL}(\rho\|\pi) + \ln(2\sqrt{n}/\delta)}{n} \right) \leq \delta, \qquad (6)$$

*where $\mathcal{P}$ is the set of all possible probability distributions on $\mathcal{H}$ that can depend on $S$.*

The following relaxation of the PAC-Bayes-kl inequality based on Refined Pinsker's relaxation of the kl divergence helps getting some intuition about the bound [McAllester, 2003]. With probability at least $1 - \delta$, for all $\rho \in \mathcal{P}$ we have

$$\mathbb{E}_{\rho}[L(h)] \leq \mathbb{E}_{\rho}[\hat{L}(h, S)] + \sqrt{2\mathbb{E}_{\rho}[\hat{L}(h, S)]\frac{\mathrm{KL}(\rho\|\pi) + \ln(2\sqrt{n}/\delta)}{n}} + \frac{2\left(\mathrm{KL}(\rho\|\pi) + \ln(2\sqrt{n}/\delta)\right)}{n}. \qquad (7)$$

If $\mathbb{E}_\rho[\hat{L}(h, S)]$ is close to zero, the middle term in the inequality above vanishes, leading to so-called "fast convergence rates" (convergence of $\mathbb{E}_\rho[\hat{L}(h, S)]$ to $\mathbb{E}_\rho[L(h)]$ at the rate of $1/n$). However, achieving low $\mathbb{E}_\rho[\hat{L}(h, S)]$ is not always possible [Dziugaite and Roy, 2017, Zhou et al., 2019]. Subsequent research in PAC-Bayesian analysis has focused on two goals: (1) achieving fast convergence rates when the variance of prediction errors is low (and not necessarily the errors themselves), and (2) reducing the $\mathrm{KL}(\rho\|\pi)$ term, which may be quite large for large hypothesis spaces. For the first goal Tolstikhin and Seldin [2013] developed the PAC-Bayes-Empirical-Bernstein inequality and Mhammedi et al. [2019] proposed to use excess losses and also derived the alternative PAC-Bayes-Unexpected-Bernstein inequality. For the second goal Ambroladze et al. [2007] suggested to use informed priors and Mhammedi et al. [2019] perfected the idea by proposing to average over "forward" and "backward" construction with informed prior. Next we explain the ideas behind the excess losses and informed priors in more details.

**Excess Losses**  Let $h^*$ be a reference prediction rule that is independent of $S$. We define the excess loss of a prediction rule $h$ with respect to the reference $h^*$ by

$$\Delta_\ell(h(X), h^*(X), Y) = \ell(h(X), Y) - \ell(h^*(X), Y).$$

If $\ell$ is the zero-one loss, the excess loss naturally gives rise to ternary random variables, but it is well-defined for any real-valued loss function. We use $\Delta_L(h, h^*) = \mathbb{E}_D[\Delta_\ell(h(X), h^*(X), Y)] = L(h) - L(h^*)$ to denote the expected excess loss of $h$ relative to $h^*$ and $\Delta_{\hat{L}}(h, h', S) = \frac{1}{|S|} \sum_{(X,Y) \in S} \Delta_\ell(h(X), h^*(X), Y) = \hat{L}(h) - \hat{L}(h^*)$ to denote the empirical excess loss of $h$ relative to $h^*$. The expected loss of a Gibbs prediction rule can then be written as

$$\mathbb{E}_\rho[L(h)] = \mathbb{E}_\rho[\Delta_L(h, h^*)] + L(h^*).$$

A bound on $\mathbb{E}_\rho[L(h)]$ can thus be decomposed into a summation of a PAC-Bayes bound on $\mathbb{E}_\rho[\Delta_L(h, h^*)]$ and a bound on $L(h^*)$. When the variance of the excess loss is small, we can use tools that exploit small variance, such as the PAC-Bayes-Empirical-Bernstein, PAC-Bayes-Unexpected-Bernstein, or PAC-Bayes-Split-kl inequalities proposed below, to achieve fast convergence rates for the excess loss. Bounding $L(h^*)$ involves just a single prediction rule and does not depend on the value of $\mathrm{KL}(\rho\|\pi)$. We note that it is essential that the variance and not just the magnitude of the excess loss is small. For example, if the excess losses primarily take values in $\{-1, 1\}$ and average out to zero, fast convergence rates are impossible.

**Informed Priors**  The idea behind informed priors is to split the data into two subsets, $S = S_1 \cup S_2$, and to use $S_1$ to learn a prior $\pi_{S_1}$, and then use it to learn a posterior on $S_2$ Ambroladze et al. [2007]. Note that since the size of $S_2$ is smaller than the size of $S$, this approach gains in having potentially smaller $\mathrm{KL}(\rho\|\pi_{S_1})$, but loses in having a smaller sample size in the denominator of the PAC-Bayes bounds. The balance between the advantage and disadvantage depends on the data: for some data sets it strengthens the bounds, but for some it weakens them. Mhammedi et al. [2019] perfected the approach by proposing to use it in the "forward" and "backward" direction and average over the two. Let $S_1$ and $S_2$ be of equal size. The "forward" part uses $S_1$ to train $\pi_{S_1}$ and then computes a posterior on $S_2$, while the "backward" part uses $S_2$ to train $\pi_{S_2}$ and then computes a posterior on $S_1$. Finally, the two posteriors are averaged with equal weight and the KL term becomes $\frac{1}{2} \left(\mathrm{KL}(\rho\|\pi_{S_1}) + \mathrm{KL}(\rho\|\pi_{S_2})\right)$. See [Mhammedi et al., 2019] for the derivation.

**Excess Losses and Informed Priors**  Excess losses and informed priors make an ideal combination. If we split $S$ into two equal parts, $S = S_1 \cup S_2$, we can use $S_1$ to train both a reference prediction rule $h_{S_1}$ and a prior $\pi_{S_1}$, and then learn a PAC-Bayes posterior on $S_2$, and the other way around. By combining the "forward" and "backward" approaches we can write

$$\mathbb{E}_\rho[L(h)] = \frac{1}{2}\mathbb{E}_\rho[\Delta_L(h, h_{S_1})] + \frac{1}{2}\mathbb{E}_\rho[\Delta_L(h, h_{S_2})] + \frac{1}{2}\left(L(h_{S_1}) + L(h_{S_2})\right), \qquad (8)$$

and we can use PAC-Bayes to bound the first term using the prior $\pi_{S_1}$ and the data in $S_2$, and to bound the second term using the prior $\pi_{S_2}$ and the data in $S_1$, and we can bound $L(h_{S_1})$ and $L(h_{S_2})$ using the "complementary" data in $S_2$ and $S_1$, respectively.

**PAC-Bayes-Empirical-Bernstein Inequalities**  The excess losses are ternary random variables taking values in $\{-1, 0, 1\}$ and, as we have already discussed, the kl inequality is not well-suited

for them. PAC-Bayesian inequalities tailored for non-binary random variables were derived by Seldin et al. [2012], Tolstikhin and Seldin [2013], Wu et al. [2021], and Mhammedi et al. [2019]. Seldin et al. [2012] derived the PAC-Bayes-Bernstein oracle bound, which assumes knowledge of the variance. Tolstikhin and Seldin [2013] made it into an empirical bound by deriving the PAC-Bayes-Empirical-Bernstein bound for the variance and plugging it into the PAC-Bayes-Bernstein bound of Seldin et al.. Wu et al. [2021] derived an oracle PAC-Bayes-Bennett inequality, which again assumes oracle knowledge of the variance, and showed that it is always at least as tight as the PAC-Bayes-Bernstein, and then also plugged in the PAC-Bayes-Empirical-Bernstein bound on the variance. Mhammedi et al. [2019] derived the PAC-Bayes-Unexpected-Bernstein inequality, which directly uses the empirical second moment. Since we have already shown that the Unexpected Bernstein inequality is tighter than the Empirical Bernstein, and since the approach of Wu et al. requires a combination of two inequalities, PAC-Bayes-Empirical-Bernstein for the variance and PAC-Bayes-Bennett for the loss, whereas the approach of Mhammedi et al. only makes a single application of PAC-Bayes-Unexpected-Bernstein, we only compare our work to the latter.

We cite the inequality of Mhammedi et al. [2019], which applies to an arbitrary loss function. We use $\tilde{\ell}$ and matching tilde-marked quantities to distinguish it from the zero-one loss $\ell$. For any $h \in \mathcal{H}$, let $\tilde{L}(h) = \mathbb{E}_D[\tilde{\ell}(h(X), Y)]$ be the expected tilde-loss of $h$, and let $\hat{\tilde{L}}(h, S) = \frac{1}{|S|} \sum_{(X,Y) \in S} \tilde{\ell}(h(X), Y)$ be the empirical tilde-loss of $h$ on a sample $S$.

**Theorem 6** (PAC-Bayes-Unexpected-Bernstein inequality [Mhammedi et al., 2019]). *Let $\tilde{\ell}(\cdot, \cdot)$ be an arbitrary loss function bounded from above by $b$ for some $b > 0$, and assume that $\hat{\tilde{\mathbb{V}}}(h, S) = \frac{1}{|S|} \sum_{(X,Y) \in S} \tilde{\ell}(h(X), Y)^2$ is finite for all $h$. Let $\psi(u) := u - \ln(1 + u)$ for $u > -1$. Then for any distribution $\pi$ on $\mathcal{H}$ that is independent of $S$, any $\gamma \in (0, 1/b)$, and any $\delta \in (0, 1)$:*

$$\mathbb{P}\left( \exists \rho \in \mathcal{P} : \mathbb{E}_\rho[\tilde{L}(h)] \geq \mathbb{E}_\rho[\hat{\tilde{L}}(h, S)] + \frac{\psi(-\gamma b)}{\gamma b^2} \mathbb{E}_\rho[\hat{\tilde{\mathbb{V}}}(h, S)] + \frac{\mathrm{KL}(\rho \| \pi) + \ln \frac{1}{\delta}}{\gamma n} \right) \leq \delta,$$

*where $\mathcal{P}$ is the set of all possible probability distributions on $\mathcal{H}$ that can depend on $S$.*

In optimization of the bound, we take the same grid of $\gamma \in \{1/(2b), \cdots, 1/(2^k b)\}$ for $k = \lceil \log_2(\sqrt{n/\ln(1/\delta)}/2) \rceil$ and a union bound over the grid, as we did for Theorem 3.

### 3.3 PAC-Bayes-Split-kl Inequality

Now we present our PAC-Bayes-Split-kl inequality. For an arbitrary loss function $\tilde{\ell}$ taking values in a $[a, b]$ interval for some $a, b \in \mathbb{R}$, let $\tilde{\ell}^+ := \max\{0, \tilde{\ell} - \mu\}$ and $\tilde{\ell}^- := \max\{0, \mu - \tilde{\ell}\}$ for some $\mu \in [a, b]$. For any $h \in \mathcal{H}$, let $\tilde{L}^+(h) = \mathbb{E}_D[\tilde{\ell}^+(h(X), Y)]$ and $\tilde{L}^-(h) = \mathbb{E}_D[\tilde{\ell}^-(h(X), Y)]$. The corresponding empirical losses are denoted by $\hat{\tilde{L}}^+(h, S) = \frac{1}{n} \sum_{i=1}^n \tilde{\ell}^+(h(X_i), Y_i)$ and $\hat{\tilde{L}}^-(h, S) = \frac{1}{n} \sum_{i=1}^n \tilde{\ell}^-(h(X_i), Y_i)$.

**Theorem 7** (PAC-Bayes-Split-kl Inequality). *Let $\tilde{\ell}(\cdot, \cdot)$ be an arbitrary loss function taking values in a $[a, b]$ interval for some $a, b \in \mathbb{R}$. Then for any distribution $\pi$ on $\mathcal{H}$ that is independent of $S$, any $\mu \in [a, b]$, and any $\delta \in (0, 1)$:*

$$\mathbb{P}\left[ \exists \rho \in \mathcal{P} : \mathbb{E}_\rho[\tilde{L}(h)] \geq \mu + (b - \mu) \, \mathrm{kl}^{-1,+}\left( \frac{\mathbb{E}_\rho[\hat{\tilde{L}}^+(h, S)]}{b - \mu}, \frac{\mathrm{KL}(\rho \| \pi) + \ln \frac{4\sqrt{n}}{\delta}}{n} \right) \right.$$
$$\left. - (\mu - a) \, \mathrm{kl}^{-1,-}\left( \frac{\mathbb{E}_\rho[\hat{\tilde{L}}^-(h, S)]}{\mu - a}, \frac{\mathrm{KL}(\rho \| \pi) + \ln \frac{4\sqrt{n}}{\delta}}{n} \right) \right] \leq \delta,$$

*where $\mathcal{P}$ is the set of all possible probability distributions on $\mathcal{H}$ that can depend on $S$.*

*Proof.* We have $\mathbb{E}_\rho[\tilde{L}(h)] = \mu + \mathbb{E}_\rho[\tilde{L}^+(h)] - \mathbb{E}_\rho[\tilde{L}^-(h)]$. Similar to the proof of Theorem 4, we take a union bound of PAC-Bayes-kl upper bound on $\mathbb{E}_\rho[\tilde{L}^+(h)]$ and PAC-Bayes-kl lower bound on $\mathbb{E}_\rho[\tilde{L}^-(h)]$. ∎

### 3.4 PAC-Bayes-split-kl with Excess Loss and Informed Prior

Looking back at the expected loss decomposition in equation (8), we can use PAC-Bayes-split-kl to bound the first two terms and a bound on the binomial tail distribution to bound the last term. For $n$ i.i.d. Bernoulli random variables $Z_1, \ldots, Z_n$ with bias $p \in (0,1)$, we define the binomial tail distribution $\mathsf{Bin}(n,k,p) = \mathbb{P}(\sum_{i=1}^n X_i \leq k)$ and its inverse $\mathsf{Bin}^{-1}(n,k,\delta) = \max\{p : p \in [0,1] \text{ and } \mathsf{Bin}(n,k,p) \geq \delta\}$. The following theorem relates $\hat{p} = \frac{1}{n}\sum_{i=1}^n Z_i$ and $p$.

**Theorem 8** (Test Set Bound [Langford, 2005]). *Let $Z_1, \ldots, Z_n$ be $n$ i.i.d. Bernoulli random variables with bias $p \in (0,1)$ and let $\hat{p} = \frac{1}{n}\sum_{i=1}^n Z_i$ be the empirical mean. Then for any $\delta \in (0,1)$:*

$$\mathbb{P}\Big(p \geq \mathsf{Bin}^{-1}(n, n\hat{p}, \delta)\Big) \leq \delta.$$

By applying Theorems 7 and 8 to equation (8) we obtain the following result.

**Theorem 9.** *For any $\mu \in [-1,1]$ and any $\delta \in (0,1)$:*

$$\mathbb{P}\left(\exists \rho \in \mathcal{P} : \mathbb{E}_\rho[L(h)] \geq \mu + (1-\mu)(a) - (\mu+1)(b) + \frac{1}{2}(c)\right) \leq \delta,$$

*where $\mathcal{P}$ is the set of all possible probability distributions on $\mathcal{H}$ that can depend on $S$,*

$$(a) = \mathrm{kl}^{-1,+}\left(\frac{1}{2}\frac{\mathbb{E}_\rho[\Delta_{\hat{L}}^+(h, h_{S_1}, S_2)]}{1-\mu} + \frac{1}{2}\frac{\mathbb{E}_\rho[\Delta_{\hat{L}}^+(h, h_{S_2}, S_1)]}{1-\mu}, \frac{\mathrm{KL}(\rho\|\pi) + \ln\frac{8\sqrt{n/2}}{\delta}}{n/2}\right),$$

$$(b) = \mathrm{kl}^{-1,-}\left(\frac{1}{2}\frac{\mathbb{E}_\rho[\Delta_{\hat{L}}^-(h, h_{S_1}, S_2)]}{\mu+1} + \frac{1}{2}\frac{\mathbb{E}_\rho[\Delta_{\hat{L}}^-(h, h_{S_2}, S_1)]}{\mu+1}, \frac{\mathrm{KL}(\rho\|\pi) + \ln\frac{8\sqrt{n/2}}{\delta}}{n/2}\right),$$

*in which $\pi = \frac{1}{2}\pi_{S_1} + \frac{1}{2}\pi_{S_2}$, and*

$$(c) = \mathsf{Bin}^{-1}\left(\frac{n}{2}, \frac{n}{2}\hat{L}(h_{S_1}, S_2), \frac{\delta}{4}\right) + \mathsf{Bin}^{-1}\left(\frac{n}{2}, \frac{n}{2}\hat{L}(h_{S_2}, S_1), \frac{\delta}{4}\right).$$

The proof is postponed to Appendix C.

## 4 Experiments

We evaluate the performance of the PAC-Bayes-split-kl inequality in linear classification and in weighted majority vote using several data sets from UCI and LibSVM repositories [Dua and Graff, 2019, Chang and Lin, 2011]. An overview of the data sets is provided in Appendix E.1. For linear classification we reproduce the experimental setup of Mhammedi et al. [2019], and for the weighted majority vote we reproduce the experimental setup of Wu et al. [2021].

### 4.1 The Experimental Setup of Mhammedi et al. [2019]: Linear Classifiers

In the first experiment we follow the experimental setup of Mhammedi et al. [2019], who consider binary classification problems with linear classifiers in $\mathbb{R}^d$ and Gaussian priors and posteriors. A classifier $h_w$ associated with a vector $w \in \mathbb{R}^d$ makes a prediction on an input $X$ by $h_w(X) = \mathbb{1}(w^\top X > 0)$. The posteriors have the form of Gaussian distributions centered at $w_S \in \mathbb{R}^d$, with covariance $\Sigma_S$ that depends on a sample $S$, $\rho = \mathcal{N}(w_S, \Sigma_S)$. The informed priors $\pi_{S_1} = \mathcal{N}(w_{S_1}, \Sigma_{S_1})$ and $\pi_{S_2} = \mathcal{N}(w_{S_2}, \Sigma_{S_2})$ are also taken to be Gaussian distributions centered at $w_{S_1}$ and $w_{S_2}$, with covariance $\Sigma_{S_1}$ and $\Sigma_{S_2}$, respectively. We take the classifier associated with $w_{S_1}$ as the reference classifier $h_{S_1}$ and the classifier associated with $w_{S_2}$ as the reference classifier $h_{S_2}$. More details on the construction are provided in Appendix E.2.

Figure 2 compares the PAC-Bayes-Unexpected-Bernstein bound PBUB and the PAC-Bayes-split-kl bound PBSkl with excess losses and informed priors. The ternary random variables in this setup take values in $\{-1, 0, 1\}$, and we select $\mu$ to be the middle value 0. Since the PAC-Bayes-kl bound (PBkl) is one of the tightest known generalization bounds, we take PBkl with informed priors as a baseline.

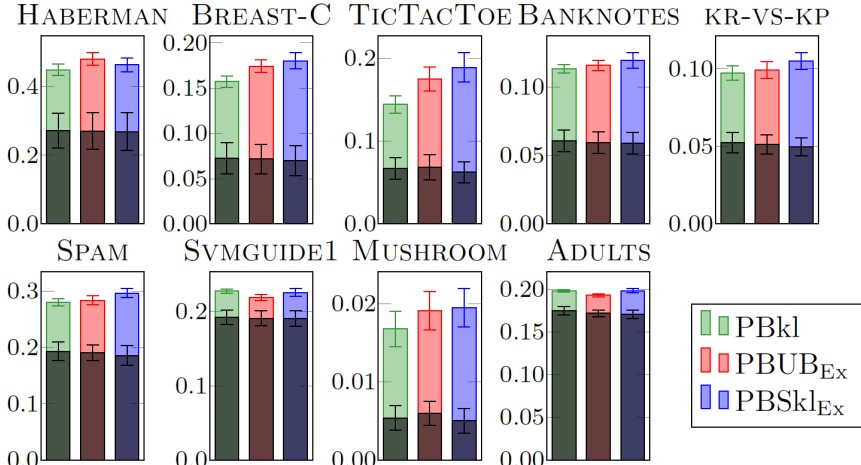

Figure 2: Comparison of the bounds and the test losses of the optimized Gaussian posterior $\rho^*$ generated by PBkl with informed priors, PBUB with excess losses and informed priors, and PBSkl with excess losses and informed priors. The test losses of the corresponding bounds are shown in black. We report the mean and the standard deviation over 20 runs of the experiments.

The details on bound calculation and optimization are provided in Appendix E.2. In this experiment all the three bounds, PBkl, PBUB, and PBSkl performed comparably. We believe that the reason is that with informed priors the $\mathrm{KL}(\rho\|\pi)$ term is small. From the relaxation of the PBkl bound in equation (7), we observe that a small $\mathrm{KL}(\rho\|\pi)$ term implies smaller difference between fast and slow convergence rates, and thus smaller advantage to bounding the excess loss instead of the raw loss. In other words, we believe that the effect of using informed priors dominates the effect of using excess losses. We note that in order to use excess losses we need to train the reference hypothesis $h^*$ on part of the data and, therefore, training an informed prior on the same data comes at no extra cost.

### 4.2 The Experimental Setup of Wu et al. [2021]: Weighted Majority Vote

In the second experiment we reproduce the experimental setup of Wu et al. [2021], who consider multiclass classification by a weighted majority vote. Given an input $X \in \mathcal{X}$, a hypothesis space $\mathcal{H}$, and a distribution $\rho$ on $\mathcal{H}$, a $\rho$-weighted majority vote classifier predicts $\mathrm{MV}_\rho(X) = \arg\max_{y \in \mathcal{Y}} \mathbb{E}_\rho[\mathbb{1}(h(X) = y)]$. One of the tightest bound for the majority vote is the tandem bound (TND) proposed by Masegosa et al. [2020], which is based on tandem losses for pairs of hypotheses, $\ell(h(X), h'(X), Y) = \mathbb{1}(h(X) \neq Y)\mathbb{1}(h'(X) \neq Y)$, and the second order Markov's inequality. Wu et al. [2021] proposed two improved forms of the bound, both based on a parametric form of the Chebyshev-Cantelli inequality. The first, CCTND, using Chebyshev-Cantelli with the tandem losses and the PAC-Bayes-kl bound for bounding the tandem losses. The second, CCPBB, using tandem losses with an offset, defined by $\ell_\alpha(h(X), h'(X), Y) = (\mathbb{1}(h(X) \neq Y) - \alpha)(\mathbb{1}(h'(X) \neq Y) - \alpha)$ for $\alpha < 0.5$, and PAC-Bayes-Empirical-Bennett inequality for bounding the tandem losses with an offset. We note that while the tandem losses are binary random variables, tandem losses with an offset are ternary random variables taking values in $\{\alpha^2, -\alpha(1 - \alpha), (1 - \alpha)^2\}$ and, therefore, application of Empirical Bernstein type inequalities makes sense. However, in the experiments of Wu et al. CCPBB lagged behind TND and CCTND. We replaced PAC-Bayes-Empirical-Bennett with PAC-Bayes-Unexpected-Bernstein (CCPBUB) and PAC-Bayes-split-kl (CCPBSkl) and showed that the weakness of CCPBB was caused by looseness of PAC-Bayes-Empirical-Bernstein, and that CCPBUB and CCPBSkl lead to tighter bounds that are competitive and sometimes outperforming TND and CCTND. For the PAC-Bayes-split-kl bound we took $\mu$ to be the middle value of the tandem loss with an offset, namely, for $\alpha \geq 0$ we took $\mu = \alpha^2$, and for $\alpha < 0$ we took $\mu = -\alpha(1 - \alpha)$.

In Figure 3 we present a comparison of the TND, CCTND, CCPBB, CCPBUB, and CCPBSkl bounds on weighted majority vote of heterogeneous classifiers (Linear Discriminant Analysis, $k$-Nearest Neighbors, Decision Tree, Logistic Regression, and Gaussian Naive Bayes), which adds the

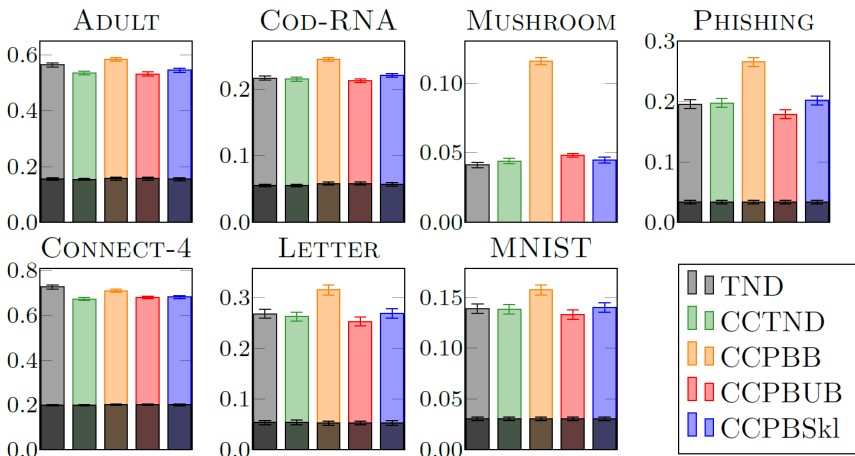

Figure 3: Comparison of the bounds and the test losses of the weighted majority vote on ensembles of heterogeneous classifiers with optimized posterior $\rho^*$ generated by TND, CCTND, CCPBB, CCPBUB, and CCPBSkl. The test losses of the corresponding bounds are shown in black. We report the mean and the standard deviation over 10 runs of the experiments.

two new bounds, CCPBUB and CCPBSkl to the experiment done by Wu et al. [2021]. A more detailed description of the experiment and results for additional data sets are provided in Appendix E.3. We note that CCPBUB and CCPBSkl consistently outperform CCPBB, demonstrating that they are more appropriate for tandem losses with an offset. The former two bounds perform comparably to TND and CCTND, which operate on tandem losses without an offset. In Appendix E.4 we replicate another experiment of Wu et al., where we use the bounds to reweigh trees in a random forest classifier. The results are similar to the results for heterogeneous classifiers.

## 5   Discussion

We have presented the split-kl and PAC-Bayes-split-kl inequalities. The inequalities answer a long-standing open question on how to exploit the structure of ternary random variables in order to provide tight concentration bounds. The proposed split-kl and PAC-Bayes-split-kl inequalities are as tight for ternary random variables, as the kl and PAC-Bayes-kl inequalities are tight for binary random variables.

In our empirical evaluation the split-kl inequality was always competitive with the kl and Unexpected Bernstein inequalities and outperformed both in certain regimes, whereas Empirical Bernstein typically lagged behind. In our experiments in the PAC-Bayesian setting the PAC-Bayes-split-kl was always comparable to PAC-Bayes-Unexpected-Bernstein, whereas PAC-Bayes-Empirical-Bennett most often lagged behind. The first two inequalities were usually comparable to PAC-Bayes-kl, although in some cases the attempt to exploit low variance did not pay off and PAC-Bayes-kl outperformed, which is also the trend observed earlier by Mhammedi et al. [2019]. To the best of our knowledge, this is the first time when the various approaches to exploitation of low variance were directly compared, and the proposed split-kl emerged as a clear winner in the basic setting, whereas in the PAC-Bayes setting in our experiments the PAC-Bayes-Unexpected-Bernstein and PAC-Bayes-split-kl were comparable, and preferable over PAC-Bayes-Empirical-Bernstein and PAC-Bayes-Empirical-Bennett.

## Acknowledgments and Disclosure of Funding

This project has received funding from European Union's Horizon 2020 research and innovation programme under the Marie Skłodowska-Curie grant agreement No 801199. The authors also acknowledge partial support by the Independent Research Fund Denmark, grant number 0135-00259B.

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
