# OpenReview forum: "Split-kl and PAC-Bayes-split-kl Inequalities for Ternary Random Variables"
_NeurIPS.cc/2022/Conference — NeurIPS 2022 Accept_

### Official Review · Reviewer_5fcJ · 2022-06-16

**Rating:** 6
**Confidence:** 5
**Soundness:** 3 good
**Presentation:** 4 excellent
**Contribution:** 2 fair

**Summary:**

The authors address the question of providing PAC-Bayes bounds for losses when the (empirical) variance is low, as previously addressed by e.g. [1, 2].

A special case of this is finding bounds for ternary losses in {-1,0,1}, which arises in two important ways:
1. bounds on the excess misclassification loss, which can also be used as per [1] to tighten PAC-Bayes bounds on the non-excess loss
2. in conjunction with the Cantelli-Chebyshev relaxation given by [3] to provide bounds on the (non-randomized) weighted majority vote via PAC-Bayes.

For losses in {0, 1} the small-kl PAC-Bayes bound [e.g. 4] is usually the tightest, even when the variance is low, but not for losses in [-1, 1] (after rescaling the bound). In order to leverage this, the authors decompose translate each random variable in the sum before decomposing it into positive and negative parts,
$$Z_i = \mu + Z_i^+ Z_i^- = \mu + \max(0, Z_i-\mu) + \max(0, -Z_i+\mu)$$
before applying the small-kl bound to the sums of $Z_i^+$ and $Z_i^-$ separately (which are both {0, 1} valued in the ternary untranslated case). This is called the *split-kl* (PAC-Bayes) bound.

This is used to prove new concentration and PAC-Bayes bounds. These are further combined with the excess risk and informed prior ideas from [1], or the Cantelli-Chebyshev relaxation from [3], and evaluated in experimental setups taken from the above.



-----

[1] Zakaria Mhammedi, Peter Grünwald, and Benjamin Guedj. PAC-Bayes un-expected Bernstein inequality.

[2] Ilya Tolstikhin and Yevgeny Seldin. PAC-Bayes-Empirical-Bernstein inequality.

[3] Yi-Shan Wu, Andres Masegosa, Stephan Lorenzen, Christian Igel, and Yevgeny Seldin. Chebyshev-cantelli pac-bayes-bennett inequality for the weighted majority vote.

[4] John Langford. Tutorial on practical prediction theory for classification.


----

UPDATE:

Overall I am not satisfied with the quite limited evaluation of this bound, which does not show clear improvements from previous results. This weakens the motivation for the paper too because of the limited number of new technical ideas.

Therefore I find myself much more on the borderline than my original review and I do agree with some of the criticisms of reviewer nL9t. However, given that related work has previously appeared at NeurIPS with similarly negligible empirical improvements, I will keep my "weak accept" score.

**Questions:**

1. Is the bound stated in Theorem 3 equivalent to the different form given by [1]? It would be nice to show this.
2. In the experiments in section 4.2, it seems all of the bounds are based on the Cantelli-Chebyshev relaxation (with the tandem bound being $\alpha = 0$). Why have you not also compared to other bounds for the weighted majority vote, in particular the first order bound $L(MV) \le 2 L(\rho)$ with the small-kl, which is often the tightest?

**Limitations:**

N/A the results are primarily of a theoretical nature.

**Strengths And Weaknesses:**

### Strengths

**Clarity and motivation**: the paper is very well written and was a pleasure to read. The relationships to previous works [1, 2] was very well explained and the incorporation of ideas from [1] was well motivated. The alternative form of the main result from [1] is an improvement in clarity to how it is stated therein and the situation of this work within its wider context was reasonably clear. My only minor criticism is that the experiments in section 4.2 do not sufficiently explain the use of the Chebyshev-Cantelli bound and majority votes as used there. This is a shame as I think the use of the split-kl bound for majority votes is a good use case.

**Relevance**: I think that the paper makes a contribution to an important and highly-active area of machine learning, improving PAC-Bayes bounds, which are among the most useful in contemporary learning theory. They bring some ideas from [1] to a wider application which is a valuable contribution.


### Weaknesses

**Technical contribution and originality**: here I think the paper falls down a bit. The main technical result is simply a decomposition of a random variable into positive and negative parts, combined with an application of the small-kl PAC-Bayes inequality. This is combined with the excess loss idea from [1] and the experimental setup therein, or the Cantelli-Chebyshev bound from [3] and their experimental setup, all of which is straightforward. Such simple ideas are can be very valuable when they lead to breakthroughs but that does not seem to be the case here, and most of the ideas used in the paper and discussed at length were originated by [1].

**Experimental results**: in the more important PAC-Bayes setting the new results are quite weak, with the new bound giving very similar results to that of [1]. The bound is not shown to be any improvement as optimization objective either. The simpler concentration inequality setting is not particularly interesting except as a motivation, and for the ternary r.v.s used an even better bound would be obtained by applying the test set bound (Th. 8) to the decomposition $Z = Z^+ - Z^-$ (i.e. a "split-Binomial" bound).

---

> ### Author Response · Authors · 2022-08-02
> **Rebuttal**
>
> We thank the reviewer for their time and feedback.
>
> “Experimental results”
>
> “... improvement as optimisation objective …”
>
> We note that PAC-Bayes-split-kl provides a computational advantage over PAC-Bayes-Unexpected-Bernstein, because the latter uses a grid of parameters $\gamma$, whereas the former has no parameters. Thus, the computation time is lowered by a multiplicative factor proportional to the size of the grid, in our experiments roughly 3-10, depending on the dataset.
>
> “Question 1. Is the bound stated in Theorem 3 equivalent to the different form given by [1]? It would be nice to show this.”
>
> Mhammedi et al. [1] presented the Unexpected Bernstein Lemma, which is a bound on the moment generating function (Lemma 10 in Appendix A in our work and Lemma 13 in [1]). Theorem 3 is a concentration of measure inequality, which follows from the Unexpected Bernstein Lemma by a standard proof technique (see the proof of Theorem 3).
>
> “Question 2. In the experiments in section 4.2, it seems all of the bounds are based on the Cantelli-Chebyshev relaxation (with the tandem bound being ). Why have you not also compared to other bounds for the weighted majority vote, in particular the first order bound  with the small-kl, which is often the tightest?”
>
> It has been shown in prior work that minimization of the first order bound deteriorates the test error of a majority vote, because it ignores correlation of errors and overconcentrates the posterior mass on the best performing classifiers [2,3]. Therefore, even though the first order bound may be tighter than the second order bound in some cases, it is not the right tool for analysing the majority vote. Masegosa et al. [3] provided an extensive comparison of the first and second order bounds and we felt that repeating it here would overload the readers, but we could add it, if the reviewers find necessary.
>
> [1] Zakaria Mhammedi, Peter Grünwald, and Benjamin Guedj. PAC-Bayes un-expected Bernstein inequality.
>
> [2] Andrés R. Masegosa, Stephan S. Lorenzen, Christian Igel, and Yevgeny Seldin. Second order PAC-Bayesian bounds for the weighted majority vote.
>
> [3] Stephan S. Lorenzen, Christian Igel, and Yevgeny Seldin. On PAC-Bayesian bounds for random forests.

---

### Official Review · Reviewer_vwKX · 2022-07-09

**Rating:** 8
**Confidence:** 4
**Soundness:** 4 excellent
**Presentation:** 4 excellent
**Contribution:** 4 excellent

**Summary:**

The paper introduces a new concentration inequality for the sum of iid bounded random variables.
The paper uses a  technique of splitting the samples with a threshold and then using a kl-inequality on each part.  This splitting allows using both the lower and upper bound kl-inequalities.
The resulting bound enjoys both the tightness of the kl-inequality and the ability to exploit the lower variance of r.v. that takes values within a segment.
The empirical comparison clearly shows how the tightness of the new split-kl bound in different regimes, compared to the empirical Berenstein and the standard kl inequalities.

The paper then derives PAC-Bayes-Split-kl inequality
and applies it to the excess loss of a binary classification problem.
The new bound exploits the lowered variance of the excess losses compared to the binary losses, and therefore, the overall split-kl-PB bound can be competitive with the standard kl-PB bound, as demonstrated on synthetic and real-world data.


**Questions:**

No additional questions

**Limitations:**

No additional limitations

**Strengths And Weaknesses:**


### Strengths
1. I believe the work is original and well-motivated.
2. The use of the splitting technique is clever and novel, as far as I know.
3. The paper is well-written and clear.
4. The authors provide an adequate survey of related work.
5. The empirical evaluation of the split-kl inequality clearly shows its merits.

### Weaknesses
1. The empirical evaluation of the split-kl-PAC-Bayes bound does not seem to give definitive conclusions, besides the looseness of PAC-Bayes-Empirical-Bennett on certain datasets.  I suggest adding more controlled synthetic experiments, as were done in Fig 1. for the concentration bounds since it can give good intuition to when certain bounds are preferable.

---

> ### Author Response · Authors · 2022-08-02
> **Rebuttal**
>
> We thank the reviewer for their time and feedback and for the suggestion to add synthetic experiments in the PAC-Bayes setup.

---

### Official Review · Reviewer_nL9t · 2022-07-10

**Rating:** 4
**Confidence:** 3
**Soundness:** 3 good
**Presentation:** 3 good
**Contribution:** 2 fair

**Summary:**

The authors introduced a new approach to a concetration inquality for random variables over a bounded interval called "split kl inequality", which first decomposes the original random variable into three terms and then applies an existing bound "kl ineqaulity" to the decomposed terms. Then the authors proposed to use the split kl inequality for PAC-Bayes bounds of generalisation error of learning alrogrithms as well as to combine it with existing approaches of excess loss and informed prior. The derived PAC Bayes generalisation error bound were compared and examined in a few different experiments.

**Questions:**

- How much is "informed prior" crucial to produce meaningful generalisation bounds in this context? — Would it be possilble to see how much improvement of bounds are done by "informed prior"?
- I was also not familier with how commonly or frequently "informed prior" is used in PAC-Bayes domain. It may be helpful to more strongly justify that "informed prior" is a reasonable approach to use in practice by additional references.
- It would be visually helpful to make clear which bound is the proposed one in Figure 2 and 3 e.g. adding "(Ours)" or something to the name lavel of the proposed one in Figure 2 and 3.
- The models in the experiments dealt in this paper seem relatively simple. I understand that proving generalisation bounds of complex models is challenging but I was personally interested in seeing if the generalisation bound still works for more complex models e.g. LeNet for MNIST.

**Limitations:**

There would not no concern for potential negative societal impact. To me personally, the current limitation is that it is difficult to interprete from experiments or equations if the proposed idea of PAC-Bayes-split-kl inequalities has imporved the generalisation bounds to a fair defree or not. For example, would the difference of number in the figures be significant in the context of PAC-Bayes? The reviewer's position on this paper is neutral and the reviewer is happy to increase the score if the technical or practical impact is well justified.


**Strengths And Weaknesses:**

The reviewer is personally very much fond of the authors' writing in this paper, which explains important matters of this work / other existing works in an intuitive and comprehensive manner. For example, the motivation of this work is nicely lined up with a proper technical level to wide audiences in introduction. In addition, the advantage of split kl inequality has been made clear in Figure 1. Comprehensive presentation and simplicity of the idea is a clear strengh of this work. My main concern is the significance / impact when we combine this idea with PAC-Bayes bounds. The derived new generalisation bound in Figure 2, 3 seemed similar to the other existing bounds at first glance, or it was unclear how to interprete the improvement level. For the first experiment for example, since the authors combined their idea of split kl inequality with existing approaches of "informed priors", some might get an impression from these figures that the "informed prior" part has already finished the majority of works to lower a bound in each bound and they may wonder about how critical the improvement by the split kl part is.

---

> ### Author Response · Authors · 2022-08-02
> **Rebuttal**
>
> We thank the reviewer for their time and feedback.
>
> “I was also not familier with how commonly or frequently "informed prior" is used in PAC-Bayes domain. It may be helpful to more strongly justify that "informed prior" is a reasonable approach to use in practice by additional references.”
>
> Informed priors were used in [1,2,3,4,5,6]. The goal of “informed priors” is to reduce the KL divergence between the posterior and the prior, which otherwise frequently dominates the PAC-Bayes bounds. This comes at the price of using some data to learn the “informed prior”, thus reducing the number of samples $n$ used for computing the bound (the remaining $n$ samples that were not used for learning the prior). Whether this price is worth paying or not depends on the data, and there are examples in both directions, as shown in the references above.
>
> “How much is "informed prior" crucial to produce meaningful generalisation bounds in this context? — Would it be possilble to see how much improvement of bounds are done by "informed prior"?”
>
> We note that “excess losses” are formed by training a reference prediction rule $h^*$ on part of the data and then computing the “excess losses” relative to the reference prediction rule. The data used for training the reference prediction rule cannot be used for computing the bound, but it can be used for constructing an “informed prior”, so it is not that “informed priors” are “crucial”, but since there is anyway data that can be used to construct them at no extra cost, it makes a lot of sense to use it.
>
> “The models in the experiments dealt in this paper seem relatively simple. I understand that proving generalisation bounds of complex models is challenging but I was personally interested in seeing if the generalisation bound still works for more complex models e.g. LeNet for MNIST.”
>
> Many previous works [1,2,3,4] studied generalization using PAC-Bayes bounds with data-dependent prior under relatively simple models. [5,6] studied the generalization of neural networks trained by specific algorithms with a different family of the data-dependent prior. Therefore, we see the potential of applying to more complex models, which we leave for future work.
>
> [1] Amiran Ambroladze, Emilio Parrado-Hernández, and John Shawe-Taylor. Tighter PAC-Bayes bounds. In Advances in Neural Information Processing Systems (NeurIPS), 2007.
>
> [2] Emilio Parrado-Hernández, Amiran Ambroladze, John Shawe-Taylor, and Shiliang Sun. PAC-Bayes bounds with data dependent priors. Journal of Machine Learning Research, 13, 2012.
>
> [3] Omar Rivasplata, Emilio Parrado-Hernandez, John Shaws-Taylor, Shiliang Sun, and Csaba Szepesvari. PAC-Bayes bounds for stable algorithms with instance-dependent priors.  In Advances in Neural Information Processing Systems (NeurIPS), 2018.
>
> [4] Zakaria Mhammedi, Peter Grünwald, and Benjamin Guedj. PAC-Bayes un-expected Bernstein inequality. In Advances in Neural Information Processing Systems (NeurIPS), 2019.
>
> [5] Gintare Karolina Dziugaite and Daniel M. Roy. Data-dependent PAC-Bayes priors via differential privacy. In Advances in Neural Information Processing Systems (NeurIPS), 2018.
>
> [6] Gintare Karolina Dziugaite, Kyle Hsu, Waseem Gharbieh, Gabriel Arpino, and Daniel M. Roy. On the role of data in PAC-Bayes bounds. In International Conference on Artificial Intelligence and Statistics (AISTATS), 2021.

---

> > ### Comment · Reviewer_nL9t · 2022-08-08
> > **Response**
> >
> > Thank you for answering the questions and clarifying the literature on informed prior. My remaining concerns is the significance of the proposed approach in the PAC-Bayes context mentioned in the Strengths And Weaknesses section. This might be due to lack of my expert knowledge on the domain but in summary I am still not fully sure how to interprete significance of the improvement in PAC-Bayes bound values demonstrated in the experiment. My score threfore has not been changed at the moment.
> >
> > As Paper2345 pointed out, substance of the new methodological idea is simple decomposition of a random variable into the positive and negative part from some centering constant. A simple approach could make a border impact if proven innovative but I would assume that it needs to be proven effective strongly given that NeurIPS is a top-tear conference.
> >
> > In the first experiment, the proposed bound seems to outperform existing bounds in many datasets. However I was not fully sure about how we should interprete the level of improvement — for example with the Haberman dataset the proposed bound seems a few % smaller than others — how much difficult/significant would it be to achieve this improvement? How should we take or interprete this percentage of improvement in this linear classifier case? In the second experiment, it was mentioned that the propose approach was “competitive” with the kl and Unexpected Bernstein inequalities and outperformed both in “certain regimes”. I was again not fully sure about how to intreprete significannce of this improvement there, although I also understand this may potentially be due to my lack of expert knowledge. My whole point was how we should justify and support the empirical significance of the proposed approach from these results?
> >
> > In Section 2, the author clarified a situation where the split kl concetration inequality works. I would personally like to see the similar detailed explanation and demonstration on in which situation the split kl PAC-Bayes bound works effectively and in which situation it does not. A proposal does not have to work in all situation because there is no free lunch, and therefore I would like to see clear demonstration of a potential situation we would clearly like to use the split kl PAC-Bayes bound. I assume that such elaboration would help justification of the significance/strength of the proposed approach.

---

> > > ### Author Response · Authors · 2022-08-08
> > > **Response**
> > >
> > > We thank the reviewer for the reply. We would like to emphasize that the contributions of our work are not confined to numerical improvements in empirical experiments. They also include the following points:
> > >
> > > 1. Design of an inequality that would simultaneously match the tightness of kl and Empirical Bernstein is a decade-old open problem, going back to the work of Mauer and Pontil (2009), who proposed the Empirical Bernstein inequality, but did not compare it to kl, on to the work of Tolstikhin and Seldin (2013), who compared PAC-Bayes-kl with PAC-Bayes-Empirical-Bernstein and showed that in some regimes one is better and in other regimes the other is better, and on to the works of Mhammedi et al. (2019), who proposed PAC-Bayes-Unexpected-Bernstein and also observed that it is sometimes tighter and sometimes weaker than PAC-Bayes-kl, and Wu et al. (2021), who proposed PAC-Bayes-Empirical-Bennett, which improved on PAC-Bayes-Empirical-Bernstein, but still had a mixed comparison with PAC-Bayes-kl. The latter three papers were published in NeurIPS. We propose the split-kl and the PAC-Bayes-split-kl inequalities and show that the base split-kl inequality is always competitive with all the alternatives, irrespective of the distribution. When it comes to PAC-Bayes and optimization of PAC-Bayes bounds, there are additional effects coming from the PAC-Bayes level, including excess losses and informed prior constructions, so the comparison is not as clear-cut as at the base level and it is challenging to give an complete guide on when it will provide the best performance and when not, because there are multiple factors involved. But we know for sure that it comes from the PAC-Bayes level, unlike prior work, where already at the base level there was no clear winner. We do not think that the hindsight simplicity of our solution can be held against us.
> > >
> > > 2. We also provide an empirical comparison of Empirical Bernstein and Unexpected Bernstein inequalities and their PAC-Bayes extensions. The two inequalities have never been compared before.
> > >
> > >
> > > [1] Andreas Maurer and Massimiliano Pontil. Empirical Bernstein bounds and sample variance penalization. In Proceedings of the Conference on Learning Theory (COLT), 2009.
> > >
> > > [2] Ilya Tolstikhin and Yevgeny Seldin. PAC-Bayes-Empirical-Bernstein inequality. In Advances in Neural Information Processing Systems (NeurIPS), 2013.
> > >
> > > [3] Zakaria Mhammedi, Peter Grünwald, and Benjamin Guedj. PAC-Bayes un-expected Bernstein inequality. In Advances in Neural Information Processing Systems (NeurIPS), 2019.
> > >
> > > [4] Yi-Shan Wu, Andres Masegosa, Stephan Lorenzen, Christian Igel, and Yevgeny Seldin. Chebyshev-cantelli pac-bayes-bennett inequality for the weighted majority vote. In Advances in Neural Information Processing Systems (NeurIPS), 2021.

---

### Official Review · Reviewer_eeea · 2022-07-11

**Rating:** 5
**Confidence:** 4
**Soundness:** 3 good
**Presentation:** 3 good
**Contribution:** 2 fair

**Summary:**

The authors present a new concentration of measure inequality for sum of independent bounded random variables namely split-kl inequality. They derive this new inequality by combining kl-inequalities (1 and 2) in a clever way. They provide empirical cmparison of this new inequalities with the existing concentration inequalities such as kl-inequality, Empirical Bernstein inequality and Unexpected Bernstein inequality. They show that their new inequality is tighter than all of these inequalities in some regimes.

They further extend their contribution to PAC Bayes setting and derive PAC-Bayes-split-kl inequality. Again, they empirically (in synthetic and real world data) identify regimes where their inequality performs better than other existing inequalities such as PAC-Bayes-kl, PAC-Bayes Empirical Bernstein, PAC-Bayes Unexpected Bernstein, and PAC-Bayes Empirical Bennett inequalities.

**Questions:**

- Can you add some experiments to show the dependency of the bound on choice of $\mu$?
- It would also be helpful to discuss the tighntness of various bounds as we increase n.


**Limitations:**

The limitations are discussed adequately.

**Strengths And Weaknesses:**

Strengths:
The paper is easy to follow and claims stem from logical arguments. The experiments are extensive and support the claims made by authors. Theoretically, the idea is simple but interestingly, it leads to good empirical results.

Weaknesses:
It is difficult to understand that how is this new inequality fundamentally different than the kl inequality. Without a careful choice of $\mu$, I am not sure if this new inequality would always be tighter than kl inequality in all the regimes. My observation comes from the following argument: consider Z $\in [a, b]$. Take $\mu = a$, then $Z^+ = Z-a$ and $Z^- = 0$. Similalry, take $\mu = b$, then $Z^+ = 0$ and $Z^- = b - Z$. In both these cases, we are just translating Z and both kl inequality and kl-split inequality should behave similar for these choices of $\mu$. Of course, there might be a clever choice of $\mu$ which makes one perform better than the other but I am not sure how to make that choice.

---

> ### Author Response · Authors · 2022-08-02
> **Rebuttal**
>
> We thank the reviewer for their time and feedback.
>
> “... I am not sure if this new inequality would always be tighter than kl inequality in all the regimes.”
>
> We do not claim that split-kl is always tighter than kl, we claim that it is never much looser, but in some cases it can be significantly tighter. Indeed, for a Bernoulli random variable it is always a bit weaker, irrespective of the choice of $\mu$, but for a random variable concentrated inside the $[a,b]$ interval and for a good choice of $\mu$, which we discuss below, it can be significantly tighter. This is also what we show in Figure 1 and Figures 4 and 5 in the supplementary material.
>
> “... a clever choice of $\mu$ …”
>
> For ternary random variables, which we work with in the paper, a natural choice is to take $\mu$ to be the middle value. With this choice both $Z^+$ and $Z^-$ are Bernoulli random variables and the kl bounds applied to $Z^+$ and $Z^-$ in split-kl are very tight. If a ternary random variable has a high probability mass on the middle value, the split-kl has the most advantage over kl. If the probability mass on the middle value is small, it means that the random variable is close to a Bernoulli random variable, and this is the regime where kl performs well, but split-kl is never much weaker. We use this value of $\mu$ in all our comparisons.
>
> For general random variables in a $[a,b]$ interval the optimal choice of $\mu$ is a challenging open question. While it may be tempting to estimate the expectation of $Z$ using part of the data and set $\mu$ to this estimate, we know that this choice is suboptimal. For example, for a ternary random variable taking values in $\{-1,0,1\}$ with zero probability mass on -1 and equal mass on 0 and 1, taking $\mu = 0$ (the middle value) is significantly better than taking $\mu = 0.5$ (the expectation). Another possibility is to take a grid of values of $\mu$ and a union bound over the grid. Since the main focus of our work was on ternary random variables, which on their own have multiple applications, we leave the study of more general random variables to future work.
>
>
> “Can you add some experiments to show the dependency of the bound on choice of $\mu$?”
>
> We tried to keep a delicate balance between demonstrating the interesting behaviour of the bounds and not flooding the reader with overly many experimental setups. We are happy to add more experiments, but we would appreciate it if the reviewer could give more details on the experimental setup they would be interested to see.
>
> “It would also be helpful to discuss the tightness of various bounds as we increase $n$.”
>
> The discussion and additional experiments are provided in Appendix D1 and Figures 4 and 5 in the supplementary material. We are happy to move some of the discussion to the body, if the reviewer finds it necessary.

---

> > ### Comment · Reviewer_eeea · 2022-08-05
> > **Response to rebuttal**
> >
> > Thank you for your detailed response.
> >
> > - My main concern remains the choice of $\mu$. As you said, choosing a good $\mu$ could be really difficult for the general case of bounded random variable in $[a, b]$. Taking a grid of values of $\mu$ and taking a union bound is an interesting idea but I believe this will have adverse effect on computation time.
> > - To show the dependency of the bound on choice of $\mu$, you can re-run your experiments with different choices of $\mu$. For example: you can choose $\mu$ to be in $\\{ -0.75, -0.5, -0.25, 0.25, 0.5, 0.75 \\}$ in Figure 1 and run the experiments again. This would, at least empirically, show how good the bounds are if we have no prior information to choose $\mu$. You can also add the bound you get by choosing a grid of values of $\mu$ and taking the union bound.
> > - Yes, Figure 4 and 5 are interesting. As $n$ increases, all the bounds get better in general (and Emprical Bernstein in particular). You could perhaps fix the value of $p_0 \in \\{ 0.1, 0.5, 0.9 \\}$ and rerun the experiments for $n \in \\{ 100, 1000, 5000, 10000 \\}$.
> > (A related comment: I believe there is a typo in the caption of Figure 7 (b))

---

> > > ### Author Response · Authors · 2022-08-08
> > > **Response**
> > >
> > > We thank the reviewer for the comments.
> > >
> > > The split-kl inequality was designed for ternary random variables, which naturally appear in multi-class classification with introduction of excess losses or in second order bounds for the weighted majority vote. And for ternary random variables we have the optimal choice of $\mu$ - it is the middle value (we explain further below). While split-kl can be applied to continuous random variables, this is not a natural application domain, nor a domain we focus on in the paper. So we do not see this concern as a weakness of the paper.
> > >
> > > We think it would be helpful to make a parallel with the kl inequality. The kl inequality is the ultimate choice for Bernoulli random variables in the sense that irrespective of the Bernoulli distribution it provides the tightest or almost the tightest concentration guarantees (well, the binomial bound is slightly tighter, but it does not combine with PAC-Bayes). Therefore, the kl inequality is a very powerful tool for classification problems. But it is not the ultimate choice for regression: depending on a distribution, in some cases it may be considerably tighter than alternative inequalities for continuous random variables, whereas in other cases it may potentially be much looser.
> > >
> > > Similarly, the split-kl inequality is the ultimate choice for ternary random variables. When we take $\mu$ to be the middle value, the kl inequalities for $Z^+$ and $Z^-$ are the tightest and we get the most out of the bound. As we demonstrate in our experiments, irrespective of the ternary distribution the split-kl is never much weaker than any alternative, and in some regimes it is much tighter.
> > >
> > > But, as the kl is not the ultimate choice for continuous random variables, the split-kl is not the ultimate choice for them either. It is somewhat better than the kl, because it allows to exploit part of the variance, but depending on whether after the split $Z^+$ and $Z^-$ are close to Bernoulli or not, it may be tighter than other alternatives or not. And it is possible to construct distributions, where for any choice of $\mu$ the resulting $Z^+$ and $Z^-$ are far from Bernoulli. We can add examples to the paper.
> > >
> > > Finally, we note that the natural application domain for split-kl is more or less the same as for kl, so we do not see the fact that it is not a natural choice for continuous random variables as a limitation of our work.

---

### Author Response · Authors · 2022-08-09
**Rebuttal Recap**

Dear Reviewers, dear Area Chair,

Thank you for the engaging discussion.

We feel that several discussion threads have focused on questions, which depart from the main focus of the paper. Therefore, we would like to take the opportunity to reiterate the main focus of the paper and delimit it from the sideline discussions.

There is the kl inequality, which is tight for Bernoulli, and there are the Empirical Bernstein and Unexpected Bernstein, which can exploit small variance. And there are the ternary random variables, which appear, in particular, with introduction of excess loss in classification and in second order bounds for the weighted majority vote. All prior inequalities are potentially loose for ternary random variables (as we demonstrate in Figure 1), and for each of the inequalities there exist distributions for which they are highly suboptimal. The question of finding a better inequality for such variables has been open for about a decade and explicitly mentioned in Tolstikhin and Seldin (NeurIPS, 2013) and Mhammedi et al. (NeurIPS, 2019).

The proposed split-kl inequality with $\mu$ set to the middle value is tight for ternary random variables (in the same sense as the kl is tight for Bernoulli), because $Z = \mu + Z^+ - Z^-$ and $Z^+$ and $Z^-$ are Bernoulli, so the kl inequalities for them are tight. For some distributions other inequalities have minor advantages, for example, if the ternary random variable happens to have a distribution close to a Bernoulli, then the kl will be slightly tighter, but the split-kl never falls far behind any alternative. This is not true for prior inequalities, because for each of them there is a distribution for which it falls far behind (see Figure 1). While *in hindsight* the solution is admittedly simple, the question has been open for about a decade and we have discussed it in depth with several leading PAC-Bayes researchers and no one told us “oh, yeah, just split the variable and you will get it”.

When we go to the PAC-Bayes level, there are several additional effects coming into play from the PAC-Bayes level. For example, the PAC-Bayes-split-kl has the $\sqrt{n}$ term under the logarithm (Foong et al. (NeurIPS, 2021) have an open question on whether it can be reduced), whereas PAC-Bayes-Unexpected-Bernstein has a union bound over the grid of $\gamma$, so sometimes the comparison goes one way, and sometimes another, but as before we can be sure that irrespective of the distribution we will never fall far behind any alternative, which is not true for prior work. And we also get some computational advantage over Empirical Bernstein and Unexpected Bernstein, because we have no parameter grids.

---

> ### Author Response · Authors · 2022-08-09
> **continued**
>
> The following points raised in the discussion are not within the main focus of the paper.
>
> Continuous distributions: while the split-kl can be applied to continuous distributions, it is not designed for them, just as the kl is not designed for them. If a continuous distribution happens to be close to ternary it will be tight, but it is possible to design distributions where for any choice of $\mu$ the resulting $Z^+$ and $Z^-$ will be far from Bernoilli and the bound may potentially be loose. We note that ternary random variables have a broad range of applications and tight inequalities for them are interesting even if they are suboptimal for continuous distributions.
>
> Comparison of the first and second order bounds for the weighted majority vote: a significant part of the difference between the first and second order bounds for the weighted majority vote comes from the difference between the first and second order *oracle* bounds. This has been studied in depth by Masegosa et al. (NeurIPS, 2020) and Wu et al. (NeurIPS, 2021). In their Figure 1, Wu et al. depict the regions, where the *oracle* first order bound is better than the *oracle* second order bound and where it is the other way around. A quote from their paper accompanying the figure: “The region below the black line, where $\mathbb E_{\rho^2} [L(h,h’)] < \mathbb E_\rho[L(h)]$, is the region of low correlation of errors. In this region the second order oracle bounds are tighter than the first order oracle bound.” [and in the region above the black line the first order oracle bound is tighter]. Therefore, there will always be situations where the first order empirical bound is tighter and situations where the second order empirical bound is tighter, no matter how tight is the PAC-Bayes bound for the oracle quantities. Nevertheless, it was shown that minimization of the first order bounds almost always increases the test error, whereas minimization of the second order bounds does not lead to this effect. Therefore, the state-of-the-art approach to analysing the weighted majority vote are the second order bounds, even if they are not always the tightest. Since the contribution of our work is not at the oracle level, it makes little sense to compare to the first order bound, since most of the difference will come from the oracle level.
>
> The natural baselines in our case are the tandem bound (TND), which uses PAC-Bayes-kl with a relaxed version of Chebyshev-Cantelli, the CCTND bound, which uses Chebyshev-Cantelli with PAC-Bayes-kl, and the CCPBB bound, which uses Chebyshev-Cantelli with PAC-Bayes-Empirical-Bennett. We introduced the CCPBUB bound, which uses Chebychev-Cantelli with PAC-Bayes-Unexpected-Bernstein and CCPBSkl, which uses Chebyshev-Cantelli with PAC-Bayes-split-kl.
>
> The two new bounds, CCPBUB and CCPBSkl, are based on the same form of the second order oracle bound as CCPBB, $L(MV_\rho)\leq \frac{\mathbb E_{\rho^2}[L_\alpha(h,h’)]}{(0.5-\alpha)^2}$. CCPBUB and CCPBSkl consistently outperform CCPBB, and are comparable among each other. To repeat the main point of our paper: the PAC-Bayes-split-kl is guaranteed to be tight for ternary random variables, in the sense that it will never be much worse than any alternative bound. And this claim is supported by the experiments. No prior bound for ternary random variables has this guarantee, and indeed the CCPBB is considerably weaker than CCPBSkl in some cases. We observed no cases where CCPBUB would be considerably weaker than CCPBSkl, and it is not trivial to construct them artificially, but we cannot exclude their existence.
>
> Regarding the comparison of our new bounds, CCPBUB and CCPBSkl, with TND and CCTND: the TND and CCTND are based on alternative ways of writing the second order oracle bound. The TND is based on a relaxation of Chebyshev-Cantelli, $L(MV_\rho)\leq 4\mathbb E_{\rho^2}[L(h,h’)]$, and CCTND is based on a different way of writing the Chebyshev-Cantelli, $L(MV_\rho)\leq \frac{\mathbb E_{\rho^2}[L(h,h’)] - 2 \alpha \mathbb E_\rho[L(h)] + \alpha^2}{(0.5-\alpha)^2}$. Both forms involve only binary losses. Therefore, application of PAC-Bayes-kl in their context is tight. Our contribution demonstrates that the alternative way of writing the second order bound proposed by Wu et al., $L(MV_\rho)\leq \frac{\mathbb E_{\rho^2}[L_\alpha(h,h’)]}{(0.5-\alpha)^2}$, leads to comparably tight empirical bounds and that the weakness of CCPBB was caused by the weakness of PAC-Bayes-Empirical-Bennett and not by a weakness of this form of the oracle bound. It also shows that we cannot hope to get much tighter bounds out of this form of the second order oracle bound, because we know that PAC-Bayes-split-kl is tight. Note that we could not make this claim when using neither PAC-Bayes-Empirical-Bennett, nor PAC-Bayes-Unexpected-Bernstein. So this is an important contribution.

---

### Meta-Review · Area_Chair_gz2D · 2022-08-26

**Recommendation:** Accept
**Confidence:** Certain

**Metareview:**

This meta review is based on the reviews, the authors rebuttal and the discussion with the reviewers, and ultimately my own judgement on the paper. There was a consensus that the paper contributes an interesting new concentration of measure inequality and derive a useful PAC-Bayes inequality. I feel this work deserves to be featured at NeurIPS and will attract interest from the community. I would like to personally invite the authors to carefully revise their manuscript to take into account the remarks and suggestions made by reviewers. Congratulations!

**Award:**

No

---

### Decision · Program_Chairs · 2022-09-14

Accept